# Federated Fine-Tuning of Vision Foundation Models via Probabilistic Masking

## Abstract

Foundation Models (FMs) have revolutionized machine learning with their adaptability and high performance across tasks; yet, their integration into Federated Learning (FL) is challenging due to substantial communication overhead from their extensive parameterization. Current communication-efficient FL strategies, such as gradient compression, reduce bitrates to around 1 bit-per-parameter (bpp). However, these approaches fail to harness the characteristics of FMs, with their large number of parameters still posing a challenge to communication efficiency, even at these bitrate regimes. In this work, we present `DeltaMask`[1], a novel method that efficiently fine-tunes FMs in FL at an ultra-low bitrate, well below 1 bpp. `DeltaMask` employs stochastic masking to detect highly effective subnetworks within FMs and leverage stochasticity and sparsity in client masks to compress updates into a compact grayscale image using probabilistic filters, deviating from traditional weight training approaches. Our comprehensive evaluations across various datasets and architectures demonstrate `DeltaMask` efficiently achieves bitrates as low as 0.09 bpp, enhancing communication efficiency while maintaining FMs performance, as measured on 8 datasets and 5 pre-trained models of various network architectures.

## 1 Introduction

Federated learning (FL) enables collaborative training of neural network models directly on edge devices (referred to as clients), locally with on-device data (Konečný et al., 2016). FL comprises multiple federated rounds, which involve server-to-client model updates dispatch, local training by clients, and server-side aggregation (e.g., via FedAvg (McMahan et al., 2017)) of clients' model updates, iteratively performed until model convergence. Despite its appealing properties for users' privacy, FL requires constant models' updates transfer between server and clients, which poses a challenge in terms of communication efficiency. This becomes even more critical when the clients are resource-constrained edge devices, which operate under limited transmission bandwidth and strict energy constraints.

Recent advances in FL have led to a variety of methods aimed at enhancing communication efficiency, particularly by reducing the data volume

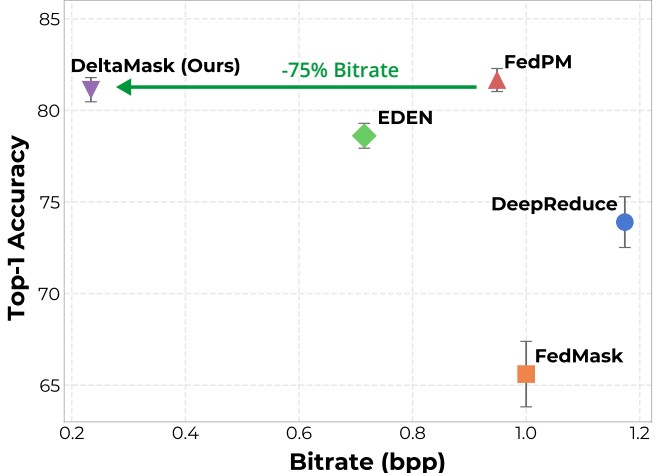

Figure 1: `DeltaMask` (Ours) vs. state-of-the-art communication-efficient FL techniques when using CLIP pre-trained ViT-B/32, results are averaged over 8 datasets.

exchanged in each federated round. These strategies often employ gradient compression techniques, including sparsification (Lin et al., 2020; Aji & Heafield, 2017),

---

[1]Code will be made available upon acceptance.

quantization (Alistarh et al., 2017; Vargaftik et al., 2022; 2021), and low-rank approximation (Mohtashami et al., 2022; Mozaffari et al., 2022), which are pivotal in streamlining data transmission.

Similarly, the "Lottery Ticket Hypothesis" (Frankle & Carbin, 2019) has paved the way for FL training regimes that diverge from traditional weight updates. Here, the focus has shifted toward identifying and cultivating high-potential subnetworks within randomly initialized neural models (Li et al., 2021a; Vallapuram et al., 2022; Li et al., 2020; Isik et al., 2023b). Such subnetworks demonstrate good performance without the need for extensive weight adjustments, offering a viable path to minimize FL communication overhead. FedMask (Li et al., 2021a) and FedPM (Isik et al., 2023b), which learn binary masks on top of random dense networks, are shown to reduce bitrate from 32 to 1 bit-per-parameter (bpp). However, jointly learning effective subnetworks in large, randomly initialized models, can severely affect training duration and model convergence. Furthermore, all aforementioned approaches primarily focus on training models from initial random weights; yet, when fine-tuning pre-trained large models, they do not exploit their unique characteristics to enable FL training at extremely low bitrates (e.g., $\ll 1$ bpp).

Leveraging the advancements in self-supervised learning, vision Foundation Models (vFMs) have brought significant advancement across various machine learning domains with their remarkable representation quality. Models like CLIP (Radford et al., 2021) and DINOv2 (Oquab et al., 2023) demonstrate rapid adaptability to diverse tasks, achieving unmatched performance in several downstream applications. Notably, recent developments have seen vision FMs, such as the ViT models, expand to billions of parameters (Dehghani et al., 2023), exemplifying the scale and complexity of modern FMs. In turn, and as an alternative to traditional fine-tuning, masking strategies have emerged (Mallya et al., 2018; Zhao et al., 2020) in a centralized setting, where selective binary masks are learned on top of frozen pre-trained weights, matching the performance of full fine-tuning. Nevertheless, the high parameter count of FMs inhibits their straightforward expansion into decentralized settings due to the substantial communication overhead (Zhuang et al., 2023), even at a bitrate of 1 bpp, thereby limiting their potential to tap into the valuable data available in distributed environments.

To bridge the gap between the high-performance potential of Foundation Models (FMs) and the practical constraints of federated settings, we introduce `DeltaMask`, an approach designed to fine-tune FMs to various downstream tasks in federated settings with significantly reduced bitrate requirements (see Figure 1). Inspired by the sparse mask updates between subsequent federated rounds, which naturally occur due to the rapid adaptability of FMs, our approach combines stochastic masking with probabilistic filters to find high-performing subnetworks within pre-trained FM, while operating in an ultra-low bitrate regime. Moreover, unlike FedPM (Isik et al., 2023b), `DeltaMask` enables an effective control over the bitrate-accuracy trade-off by simply adjusting the bits-per-element (bpe) in probabilistic filters. This paves the way for fine-tuning FMs in federated settings without the massive communication burden caused by their large number of parameters, crucial for scenarios where niche, sensitive data must remain local yet is immensely valuable for learning, such as in healthcare applications. Concisely, the main contributions of our work are as follows:

- We present a simple, yet effective, method termed `DeltaMask`, to allow fine-tuning of FMs in federated settings in a highly communication-efficient manner.
- We combine stochastic binary masks with probabilistic filters to compactly communicate mask updates and reduce bitrate bpp below 0.1.
- Our evaluation across 8 datasets and 5 pre-trained models of various network architectures demonstrate strong effectiveness of `DeltaMask` to fine-tune FMs compared to existing FL techniques.

## 2 Related Work

**Communication-Efficient FL.** Enhancing communication efficiency in FL can be achieved through fast adaptation to downstream tasks by utilizing methods such as adaptive optimizers (Reddi et al., 2021) or efficient client-sampling processes (Chen et al., 2022) that accelerate the convergence rate and, consequently, minimize the data transmission requirement. Alternatively, model updates can be compressed using gradient compression strategies like sparsification (Lin et al., 2020; Aji & Heafield, 2017), quantization (Vargaftik et al., 2022; 2021; Kostopoulou et al., 2021), and low-rank approximation (Mohtashami et al., 2022; Mozaffari et al., 2022), which reduce the volume of data transmitted during the learning process by shrinking the

size of updates communicated in each round. Likewise, training masks over densely initialized networks has been explored to provide communication efficiency between client and server (Isik et al., 2023b; Vallapuram et al., 2022). In HideNseek (Vallapuram et al., 2022) binary masks were extracted using the piece-wise sign function over client's mask scores prior to transmission. Alternatively, in FedPM (Isik et al., 2023b), stochastic mask training was utilized to sample binary masks from locally trained mask probabilities using a Bernoulli distribution, which were then transmitted to the server.

To reduce the bitrate below 1 bpp, FedPM (Isik et al., 2023b) employs arithmetic coding (Rissanen & Langdon, 1979) to encode masks based on the sparsity level (frequency of activations). However, arithmetic coding is characterized by computational complexity and intricate implementation, while its efficacy is limited due to the stochasticity of mask sampling, which leads to inconsistent mask activations. In contrast, our approach improves stochastic mask training over pre-trained FMs, introducing a novel lightweight communication protocol that employs probabilistic filters by leveraging the sparsity in subsequent mask changes. Alongside (Isik et al., 2023b), we align `DeltaMask` with gradient compression baselines, such as EDEN (Vargaftik et al., 2022) and DeepReduce (Kostopoulou et al., 2021), which operate in a similar bitrate regime ($\approx 1$ bpp).

**Masking Neural Networks.** Masks learned using gradient descent on pre-trained models can yield subnetworks with performance in parallel to conventional fine-tuning for various downstream tasks, as first demonstrated by(Mallya et al., 2018). A similar observation was evident in (Zhao et al., 2020), where masking was utilized on language models. Following the Lottery Ticket Hypothesis (Frankle & Carbin, 2019), the same idea was expanded to randomly initialized densely connected networks (Zhou et al., 2020; Ramanujan et al., 2020; Aladago & Torresani, 2021), where mask training was shown to act as an alternative form of weight training. These concepts were applied in FL in (Li et al., 2021a; Vallapuram et al., 2022; Isik et al., 2023b) to deal with different challenges in FL, such as personalization, communication efficiency and privacy. Compared to these works, our masking method focus on pre-trained foundation models and further reduce the required bitrate below the 1 bpp threshold by communicating the minimal information to reconstruct the clients' masks through probabilistic filters (Graf & Lemire, 2022).

## 3 Methodology

### 3.1 Preliminaries

In this section, we introduce fundamental concepts utilized in `DeltaMask`: (i) probabilistic filters for efficient encoding of clients' mask updates, and (ii) stochastic mask training, employed to train probabilistic masks on distributed clients and aggregate them on server-side.

**Probabilistic Filters.** Probabilistic filters are data structures that map a universe of keys, denoted as $\mathcal{U}$, of varying bit lengths, to fixed-size bit values, thereby compacting real-world data representations effectively. They achieve this by using hash functions to transform and store data in a uniformly distributed array, known as the fingerprints $\mathcal{H}$. This compact representation $\mathcal{H}$ facilitates efficient membership checking, with an adjustable rate of false positives — where a non-member might be incorrectly identified as a member — while ensuring zero false negatives. There are multiple variations of probabilistic filters, we focus on *binary fuse filters* (BFuse) (Graf & Lemire, 2022), which are known for their exceptional space efficiency and computational effectiveness. These filters offer a space efficiency of 8.62 bits per entry and a low false positive rate (up to $2^{-32}$).

Formally, an $\mu$-wise BFuse utilizes $m$ distinct hash functions $h_j : \{0, 1, \ldots, 2^n - 1\} \to \{1, 2, \ldots, l\}$, for $j = 1, \ldots, \mu$, where $l$ denotes the size of the fingerprints array, $\mathcal{H}$. Let $f : \mathbb{N} \to \{0, 1, \ldots, 2^n - 1\}$ be the fingerprint generation function, mapping each key to an $n$-bit value. For a given set of keys $\mathcal{U}$, we can compute the fingerprint array $\mathcal{H}$ as:

$$\mathcal{H} = \bigcup_{k \in \mathcal{U}} \phi(k) = \bigcup_{k \in \mathcal{U}} \left( \bigcup_{j=1}^{m} \{h_j(f(k))\} \right) \tag{1}$$

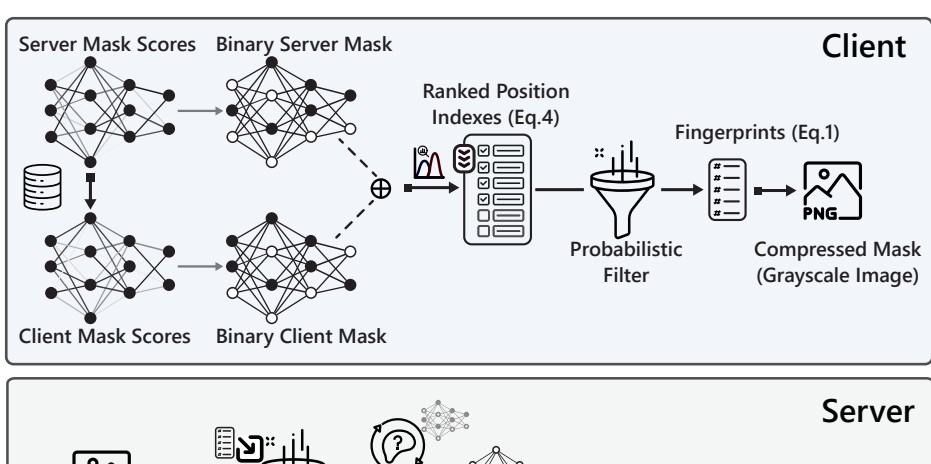

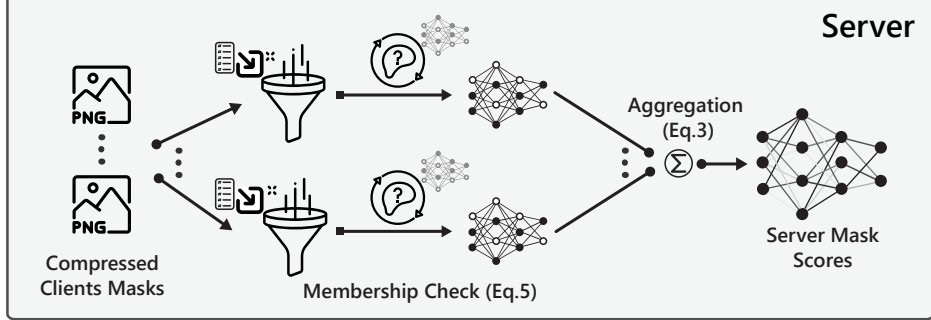

Figure 2: Overview of `DeltaMask`: Discrepancies between received and learned masks are ranked using relative entropy. Essential updates are selected via $top_\kappa$, hashed, and compressed into a grayscale image. The server reconstructs clients' masks through membership queries and updates the global mask using Bayesian aggregation.

Here, $\phi(k)$ computes the set of $m$ locations in $\mathcal{H}$ for each key $k$ in $\mathcal{U}$. Once $\mathcal{H}$ is constructed, we can perform a membership check as:

$$\text{Member}(x) = \begin{cases} \text{true,} & \bigoplus_{j=1}^{m} \mathcal{H}\left[h_j(f(x))\right] = f(x) \\ \text{false,} & \text{otherwise} \end{cases} \tag{2}$$

where, $\bigoplus_{j=1}^{m} \mathcal{H}[\cdot]$ represents the bitwise XOR operation performed on the array values of $\mathcal{H}$, indicated by the hash functions $h_j(f(x))$. The *Member*($\cdot$) function returns true if the result of the XOR operation over $\mathcal{H}$ matches the fingerprint $f(x)$, suggesting that $x$ is likely a member of the set, and false in all other occasions. Note that while computing a large number of hashes may seem daunting, not all hashing algorithms are computationally intensive. For example, BFuse use MurmurHash3 (Appleby, 2016), which is computationally efficient and exhibits exceptional properties for hashing large data structures into space-efficient arrays (e.g., uniform hash distribution and randomness).

**Stochastic Mask Training.** Unlike the thresholding mechanisms (Li et al., 2021a; Vallapuram et al., 2022; Mozaffari et al., 2022) that creates binary masks by clipping mask scores $s \in \mathbb{R}^d$, stochastic mask training (Isik et al., 2023b), involves drawing a binary mask $m \in \{0,1\}^d$ from the underlying mask's probability $\theta$ using the Bernoulli distribution (noted as $m \sim \text{Bern}(\theta)$). To generate $\theta$ from the *unbounded* mask scores $s$, a sigmoid transformation is applied (i.e., $\theta = \text{Sigmoid}(s)$). Hence, $m$ is used during the forward pass to compute the loss $\mathcal{L}(\cdot)$, and $\theta$ is subsequently updated through back-propagation. As the values of $s$ remain *unbounded*, it allows for an unbiased estimation of the true aggregate of the local clients' mask probabilities through Bayesian aggregation (Ferreira et al., 2021). Specifically, it refines the global model at round $t$ in federated setting by treating the stochastic mask's probability $\theta^{g,t}$ as a Beta distribution $\text{Beta}(\alpha_{g,t}, \beta_{g,t})$, with $\alpha_{g,t}$ and $\beta_{g,t}$ initialized to $\lambda_0$. These parameters are updated with the aggregated local binary masks from participating clients (denoted as $\bar{\theta}^{g,t}$), computed as $\alpha_{g,t} = \alpha_{g,t-1} + \bar{\theta}^{g,t}$ and $\beta_{g,t} = \beta_{g,t-1} + K \cdot \mathbf{1}_d - \bar{\theta}^{g,t}$. The aggregated probability mask is then calculated by:

$$\theta^{g,t} = \frac{\alpha_{g,t} - 1}{\alpha_{g,t} + \beta_{g,t} - 2}, \tag{3}$$

where the division is performed element-wise division. For best performance, $\alpha$ and $\beta$ are periodically reset to $\lambda_0 = 1$ at a rate inverse to the participation rate $p$ (Isik et al., 2023b). It's important to note that while the model's weight values remain unchanged, the binary mask $m$ selectively activates neurons by element-wise multiplication with the initialized model weights $w_{\text{init}}$, denoted as $w_{\text{k},t} = m^{\text{k},t} \odot w_{\text{init}}$.

### 3.2 Federated Masked Fine Tuning of Self-Supervised Foundation Models

We now present the `DeltaMask` training pipeline (see Figure 2). First, clients initialize a neural network $p_{w_{\text{init}}}$ with the weight vector $w_{\text{init}} = (w_{\text{init},1}, w_{\text{init},2}, \ldots, w_{\text{init},d}) \in \mathbb{R}^d$, from the pre-trained foundation model. The weight vector $w_{\text{init}}$ is kept fixed and never modified during training. `DeltaMask` collaboratively learns a probabilistic mask $\theta \in [0,1]^d$ using stochastic mask training, such that the function $\mathcal{L}_{\dot{W}}$ minimize its error rate on a given downstream task ($\dot{W} = m \odot w_{\text{init}}$). Specifically, at every federated round $t$, the server samples a set $K_t$ participants ($|K_t|=K$ out of $N$ clients), which individually train their local probability masks $\theta^{\text{k},t}$ on their locally stored datasets $D^{\text{k}}$, each composed of $|D_{\text{k}}|$ samples.

Instead of communicating the stochastic binary mask $m$, we significantly reduce the required bits-per-parameter (bpp) during the training process by solely communicating the subsequent key updates (indicated as position indexes set $\Delta$) between the received and trained mask. In particular, we efficiently represent $\Delta$ using probabilistic filters and transmit the fingerprint set $\mathcal{H}$ (see Eq. 1) to the server via means of a single gray-scaled image. On server-side reconstruction of clients masks $m^{\text{k},t}$ is feasible via fast membership check using the probabilistic filter (see Eq. 2). These local masks are then aggregated by the server (i.e., by using Eq. 3) to complete the $t_{th}$ round.

**Compressing Mask Updates.** Our approach utilizes a local training scheme for probability masks, where clients aim to learn a binary mask via stochastic mask training. In brief, clients receive a global probability mask $\theta^{\text{g},t-1}$ at round $t$, where each client k performs local training and updates the mask via back-propagation. To satisfy $\theta^{\text{k},t} \in [0,1]^d$ without clipping, we apply a sigmoid operation over the mask's *unbounded* mask scores $s \in \mathbb{R}^d$. Then, clients can utilize a binary mask $m^{\text{k},t}$, i.e, sampled from $\text{Bern}(\theta^{\text{k},t})$ and aim to minimize $\mathcal{L}(p_{w_{\text{k},t}}, D^{\text{k}})$ over their locally stored data, $D^{\text{k}}$, after which they back-propagate to update their mask scores.

To enable ultra-low bitrate levels, `DeltaMask` leverages the inherent sparsity in consecutive mask updates in subsequent federated rounds. For a given round $t$, we deterministically sample a binary mask $m^{\text{g},t-1}$ from the received mask distribution, $\text{Bern}(\theta^{\text{g},t-1})$ using a publicly shared *seed*. This ensures uniformity of the generated binary mask among all clients ($m_i^{\text{g},t-1} = m_j^{\text{g},t-1}$ for any $i,j \in K$). Instead of communicating $m^{\text{k},t}$ (or a compressed version of it), we catalog the index positions of differences between $m^{\text{g},t-1}$ and $m^{\text{k},t}$ to create a set of index differences, denoted as $\Delta^{\text{k},t}$. As the mask updates sparsity progressively increase during training, we introduce a $top_\kappa$ ranking that selects $\kappa\%$ of $\Delta^{\text{k},t}$ based on their relative entropy between the received and the updated probability masks. As a result, we bring a notion of importance sampling into the communication scheme, similar to (Isik et al., 2023a; Chatterjee & Diaconis, 2017; Havasi et al., 2018), helping minimize the distributed mean estimation error under low bitrate regimes. It provides essential updates early in training and conveys detailed information of $m^{\text{k},t}$ without significantly increasing bitrate due to the increasing sparsity of mask differences. Using Kullback–Leibler (KL) divergence as a measure of entropy, $\Delta'^{\text{k},t}$ is formally defined as:

$$\Delta'^{\text{k},t} = \underset{\text{KL}(\theta^{\text{k},t}, \theta^{\text{g},t-1})}{Sort} \left\{ i \mid m_i^{\text{g},t-1} \neq m_i^{\text{k},t}, \forall i \in d \right\} [1 : \mathsf{K}], \tag{4}$$

where $d$ is the dimension of the probability mask $m$, and $\mathsf{K}$ represents the number of elements to be retained in the sorted set, determined as $\kappa\%$ of $|D_{\text{k}}|$.

Next, we utilize an 4-wise *binary fuse filters* with 8-bit per entry (noted as `BFuse8`) to extract a fingerprint array $\mathcal{H}^{\text{k},t}$ from $\Delta'^{\text{k},t}$, following Eq. 1. By doing so, we essentially transition from 32-bit indexes to $\approx$ 8-bit hashed entries. In comparison with other probabilistic data structures (e.g., Bloom filters), *binary fuse*

*filters* offer an exceptional balance between space efficiency and false positive rate, while their considerable computational benefits make them ideal for resource-constrained devices often used in FL. We further encode the fingerprints set $\mathcal{H}^{k,t}$ into a pseudo gray-scale image using lossless image compression techniques $\Psi(\cdot)$. Specifically, we employ PNG-like compression (i.e., DEFLATE) due to its wide optimization for edge devices — both the compression algorithm itself and the associated transmission techniques. This approach leverages possible non-uniform distributions of entries across the fingerprints locations to further reduce the bitrate. Note that while other lossless compression schemes could be used here; yet, their impact on bitrate is minor since the primary reduction comes from transitioning from $\Delta'^{k,t}$ to $\mathcal{H}^{k,t}$. The resulting image, denoted as $A_{k,t}$, now efficiently encapsulates the mask updates in a visual and compressed format, suitable for transmission to the server.

**Bayesian Aggregation of Compressed Masks.** Once the local training at round $t$ is completed, the server needs to form a new global probability mask from the received clients' gray-scale images, $A_{k,t}$. Specifically, for each client k, server first decompress the gray-scale image to the extract $\mathcal{H}^{k,t}$ using $\Psi^{-1}(A_{k,t})$ and reconstruct the client's *original* probabilistic filter, $\texttt{BFuse8}_k$. The indexes of updates from prior server mask $m^{g,t-1}$ for client k can now be estimated via a membership query across all possible indexes of $m_{g,t-1}$, as follows:

$$\hat{\Delta}'^{k,t} = \{i \mid \text{Member}\,(i) = \text{true}, \forall i \in d\} \tag{5}$$

The clients' stochastic binary sample mask $m^{k,t}$ from $\text{Bern}(\theta^{k,t})$ can be constructed by a simple "*bit-flip*" of $m^{g,t-1}$ in the positions derived from $\hat{\Delta}'^{k,t}$. Now, server can compute the estimated aggregated probability mask $\bar{\theta}^{g,t} = \frac{1}{K}\sum_{k \in K} m^{k,t}$, which is an unbiased estimation of the underlying probability mask $\theta^{g,t} = \frac{1}{K}\sum_{k \in K} \theta^{k,t}$ using (Ferreira et al., 2021) or a similar strategy. Furthermore, $\texttt{DeltaMask}$ has a bounded estimation error (proof given in Appendix B) defined as:

$$\mathbb{E}_{M^{k,t} \sim \text{Bern}(\theta^{k,t}) \forall k \in K} \left[ \left\| \theta^{g,t} - \bar{\theta}^{g,t} \right\|_2^2 \right] \leq \frac{d}{4K} \tag{6}$$

In contrast to related approaches, such as FedPM (Isik et al., 2023b) and HideNseek (Vallapuram et al., 2022), our method enables better control over the model generalization versus bitrate bpp during FL training stage. Specifically, HideNseek transmits raw mask updates, and FedPM uses arithmetic encoding to further reduce bitrate; thus, both methods' control over the bitrate vs. model generalization trade-off depends on the clients' mask structure (i.e., the frequency of ones and zeros). In contrast, $\texttt{DeltaMask}$ allows finer control of this trade-off by adjusting $\kappa$ and bits per element (bpe) of probabilistic filters. While we primarily focus on 8-bit per entry (*bpe*) in our probabilistic filters, we provide analysis of *bpe* in Section 5.5. We also provide a complete algorithm in Algorithm 1.

### 3.3 Weight Initialization

The neural network $p_{w_{\text{init}}}$ is initialized using weights $w_{\text{init}} = (w_{\text{init},1}, w_{\text{init},2}, \ldots, w_{\text{init},d}) \in \mathbb{R}^d$ derived from a pre-trained foundation model, yet, the classification head for downstream tasks is randomly initialized. This means that while the pre-trained backbone offers high-quality features useful across various tasks, the randomly initialized classifier head significantly influences the model's overall performance. Prior research has sampled weights from a uniform distribution around the Kaiming initialization to find highly-performing subnetworks on randomly initialized network (Isik et al., 2023b; Zhou et al., 2020; Ramanujan et al., 2020; Zhou et al., 2020). However, as we focus on pre-trained models, we allow the classification head to adapt during a single round of linear probing, where the rest of the model remains frozen. This yields more stable results and rapid convergence. For a fair comparison, we employ identical weights initialization methods across all considered baselines. We also investigate scenarios with extremely low bitrates, where, linear probing is not feasible in Appendix C.6.

---

**Algorithm 1** `DeltaMask` algorithm. We provide the Bayesian Aggregation of compressed masks in Appendix A.

---

1: Server initialize global model $G$ with pretrained model weights $w_{\text{init}}$.
2: Server initialize mask weights $\theta^{g,0} \in \mathbf{R}^d$ and Beta priors $\alpha^{g,0}=\beta^{g,0}=\lambda_0$.
3: **for** $r = 1, \ldots, R$ **do**
4:     Randomly select $K$ clients to participate in round $t$
5:     **for** each client $k \in K$ **in parallel do**
6:        Sample binary server mask $m^{g,t-1} \sim \text{Bern}_{t-1}(\theta^{g,t-1})$
7:        $\theta^{k,t} \leftarrow \text{ClientUpdate}(\theta^{g,t-1})$
8:        Sample binary mask $m^{k,t} \sim \text{Bern}(\theta^{k,t})$
9:        $\Delta'^{k,t} \leftarrow \text{Sort } \{i \,|\, m^{k,t} \neq m^{g,t-1}\}_{i \in d} \, [1:\mathsf{K}]$             ▷ *// See Equation 4*
10:        $\mathcal{H}^{k,t} \leftarrow \bigcup_{i \in \Delta'^{k,t}} \phi(i)$                        ▷ *// See Equation 1*
11:        $\text{PNG}^{k,t} \leftarrow \Psi(\mathcal{H}^{k,t})$
12:     **end for**
13:     **for** each client $k \in K$ **do**
14:        $\mathcal{H}^{k,t} \leftarrow \Psi^{-1}(\text{PNG}^{k,t})$
15:        $\hat{\Delta}'^{k,t} \leftarrow \{i \,|\, \text{Member}(i) = \text{true}\}_{i \in d}$             ▷ *// See Equation 5*
16:        $m^{k,t} \leftarrow m^{g,t-1}\text{XOR } \mathcal{F}$           ▷ *// $\mathcal{F}$ is 1 in all positions of $\hat{\Delta}'^{k,t}$ and 0 otherwise*
17:     **end for**
18:     $\theta^{g,t} \leftarrow \text{BayesAgg}(\{m^{k,t}\}_{k \in K}, t, \rho)$
19: **end for**
20:
21: **procedure** CLIENTUPDATE($\theta$)
22:     **for** epoch $e = 1, 2, \ldots, E$ **do**
23:        **for** batch $b \in \mathcal{D}^k$ **do**
24:           Sample a binary mask $m \sim \text{Bern}(\theta)$
25:           $\theta \leftarrow \theta - \eta \nabla_\theta \left( \mathcal{L}_{CE} \left( y, p_{m \odot w_{\text{init}}} (y|x_b) \right) \right)$
26:        **end for**
27:     **end for**
28:     **return** $\theta$
29: **end procedure**

---

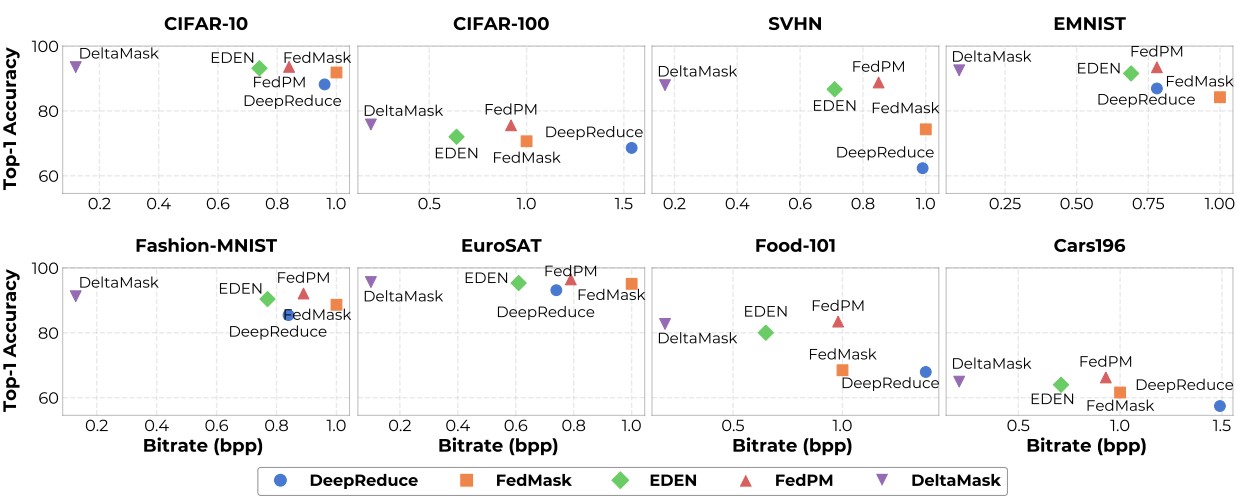

Figure 3: Performance evaluation of `DeltaMask (Ours)` in terms of average bitrate (bits-per-parameter) during FL training using $Dir(10)$ over classes ($C_p \approx 1.0$ / **IID settings**) for CLIP ViT-B/32. Federated parameters are set to $N$=30, $R$=100, $\rho$=1, and $E$=1. Detailed performance metrics, including comparison with Linear Probing and Fine-tuning, can be found in Table 2 of Appendix C.2

# 4  Experiment Setup

**Datasets and Models.** We conduct experiments across 8 diverse image classification datasets, namely CIFAR-10 (Krizhevsky, 2009), CIFAR-100 (Krizhevsky, 2009), SVHN (Netzer et al., 2011), EMNIST (Cohen et al., 2017), Fashion-MNIST (Xiao et al., 2017), EuroSAT (Helber et al., 2017), Food-101 (Bossard et al., 2014), and Cars196 (Krause et al., 2013). Here, we solely focus on classification tasks, enabling direct comparisons with similar approaches (Kostopoulou et al., 2021; Isik et al., 2023b; Li et al., 2021a; Vargaftik

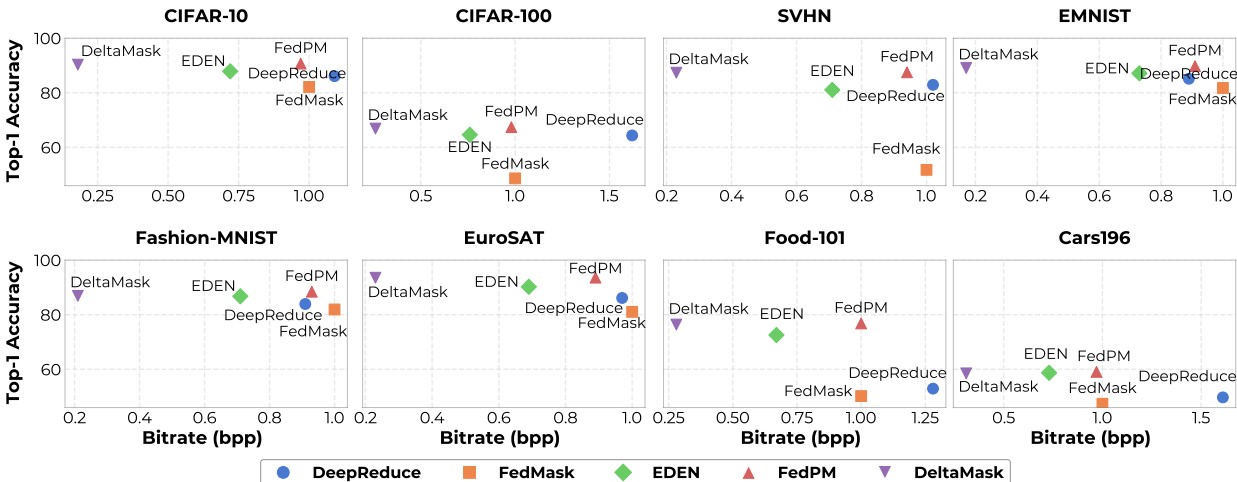

Figure 4: Performance evaluation of `DeltaMask` **(Ours)** in terms of average bitrate (bits-per-parameter) during FL training using $Dir(0.1)$ over classes ($C_p \approx 0.2$ / **non-IID settings**) for CLIP ViT-B/32. Federated parameters are set to $N$=30, $R$=300, $\rho$=0.2, and $E$=1. Detailed performance metrics, including comparison with Linear Probing and Fine-tuning, can be found in Table 3 of Appendix C.3

et al., 2022; 2021); however it is crucial to note that `DeltaMask` does not impose restrictions on the underlying downstream task.

We utilize popular ViT architectures pre-trained in self-supervised manner, such as CLIP (Radford et al., 2021), DINOv2 (Oquab et al., 2023), where, we learn a mask for the last five transformer blocks and keep all prior blocks frozen, similar to (Zhao et al., 2020). In all experiments with CLIP, we use CLIP ViT-B/32, unless stated otherwise. Apart from ViT architectures, we also evaluate convolution-based model, namely ConvMixer-768/32 (Trockman & Kolter, 2022), where mask is learned for the last five convolutional blocks, similar to ViT. We use pre-trained weights from HuggingFace for each backbone architecture and report results averaged over three runs.

**FL Setup.** We performed experiments using Flower (Beutel et al., 2020). In our data splitting procedure, we utilized Dirichlet distribution over classes, denoted as $Dir(a)$, where $a$ refers to the distribution concentration, following (Li et al., 2021b). For IID experiments, we fix $a$=10, essentially setting the class distribution per client (referred to as $C_p$) to $\approx 1.0$ (examples from all classes are available across clients), while for non-IID settings, we set $a$=0.1, which results in a $C_p \approx 0.2$. In each round, clients perform a single local training step ($E$=1) and we use a cosine scheduler for $top_\kappa$ mechanism starting from $\kappa$=0.8. Due to limited space, we provide additional details of the experimental setup in Appendix C.1.

**Baselines.** We evaluate `DeltaMask` in terms of accuracy, bitrate (bits-per-parameters), computational complexity and total volume of communicated data between client and server. As baselines, we include both fine-tuning and linear probing, where the latter involves training a classifier on top of a frozen backbone architecture. This establishes an upper performance bound for all remaining baselines. From the domain of gradient compression techniques, we incorporate EDEN (Vargaftik et al., 2022), DRIVE (Vargaftik et al., 2021), QSGD (Alistarh et al., 2017), and FedCode (Khalilian et al., 2023) into our evaluation, which operate in a close bitrate regime ($\approx 1$ bpp). Additionally, we consider DeepReduce (Kostopoulou et al., 2021) as a baseline owing to its analogous use of bloom filter-based compressor — enabling a direct comparison with BFuse filters performance. From masking strategies within FL, we assess `DeltaMask` by comparing it against the threshold-based FedMask (without initial pruning) and the stochastic masking FedPM. We use a fixed number of rounds across all baselines to facilitate a direct comparison of data transfer volumes for fine-tuning FMs, as inferred from the reported bitrates (lower is better).

# 5 Results

## 5.1 Main Empirical Findings

For ease of reading, we provide a concise list of our empirical findings below, each linked to its related subsection.

- **Section 5.2**: `DeltaMask` offers significant bitrate reductions — up to 9-fold and 6-fold decreases in IID and non-IID settings, respectively — while maintaining accuracy in par to prior approaches.

- **Section 5.3**: `DeltaMask` excels in environments with strict data transmission limits, achieving the highest accuracy per total bit transmitted compared to all considered methods. Furthermore, its computational efficiency rivals traditional gradient compression schemes, exhibiting similar encode and decode times.

- **Section 5.4**: Our approach is model-agnostic, effective across various architectures, and performs exceptionally well with large models, such as ViT-L/14.

- **Section 5.5**: By adjusting $\kappa$ and the *bpe* of filters, `DeltaMask` enables a simple, yet effective control over bitrate versus model generalization trade-off.

## 5.2 Bitrate-Accuracy Trade-Off

**IID Data Split.** Here, we focus on IID data distribution, with the number of clients ($N$) set to 30 and full client participation ($\rho$=1). As depicted in Figure 3, `DeltaMask` achieves significant reductions in communication costs compared to the considered baselines - consistently across all datasets. Among the baselines, EDEN requires the least bandwidth, while FedPM attains the highest accuracy; nevertheless, `DeltaMask` reliably matches the accuracy of FedPM. This notable improvement in bitrate over FedPM indicates that mask updates entail significant overhead. Transmitting only essential information via binary fuse filters leads to considerable reductions in bpp (up to approximately $9\times$ less) without compromising on model accuracy. Compared to DeepReduce — which utilizes Bloom filters to transmit the updates — our method underscores the importance of accurate mask reconstruction, as Bloom filters are prone to a higher false positive rate for the same number of hash functions and bits per entry. Further experimental results under conditions of low client participation in IID settings are presented in Appendix C.2.

**Non-IID Data Split.** We now evaluate in a more realistic federated setting, where clients' data follow a non-IID distribution using $Dir(0.1)$ over classes ($C_p \approx 0.2$). Furthermore, the number of clients ($N$) set to 30, while we consider partial participation with $\rho$=0.2 (meaning that in each round $\rho \cdot N = 6$ clients are randomly selected). This represents a challenging and realistic scenario, where clients have limited network resources and their data generation processes differ drastically, leading to different data distribution. From Figure 4, we notice similar gains over the baselines as the IID experiments presented in Figure 3; in that `DeltaMask` can maintain a learning procedure that results in better generalizable model, despite having up to a 6-fold reduction in the communication cost. Furthermore, the Bayesian aggregation mechanism, as presented in Eq. 3, is pivotal in achieving high accuracy when $\rho < 1$ under non-IID settings; evident from the increase in performance of DeepReduce compared to FedMask that performs poorly across all datasets. Note that in IID setting with full client participation, this behavior was reversed. We provide additional experiments under a similar non-IID setting with full client participation in Appendix C.3, where the superiority of `DeltaMask` in terms of bitrate reduction is evident.

## 5.3 Data Volume & Computational Cost Improvements

We evaluate the impact of `DeltaMask` on computational resources at both client and server levels, as well as the communication efficiency relative to the total data volume transmitted. For this, we perform experiments using CLIP on CIFAR-100 with $N$=10, and measure the encoding and decoding times for various gradient compression schemes: EDEN, DRIVE, and FedCode. Additionally, DeepReduce is included for comparison with `DeltaMask` in terms of efficiency against Bloom-based compression. From masking approaches, we

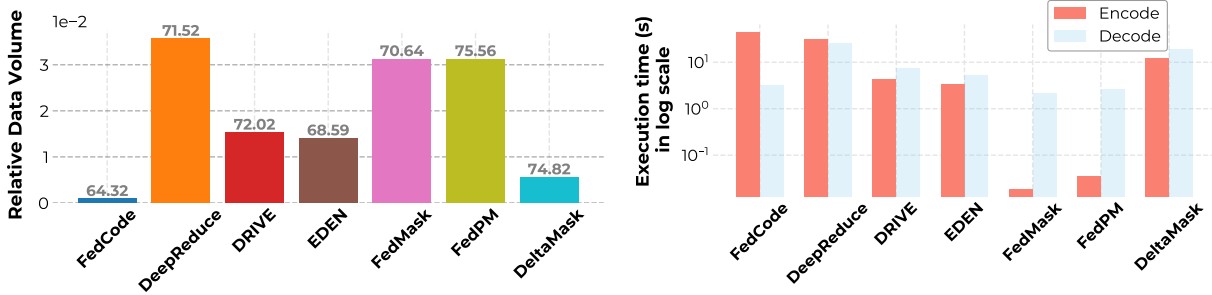

(a) Relative Data Volume against fine-tuning size.     (b) Encode/Decode run time on CPU.

Figure 5: Performance evaluation of **DeltaMask (Ours)** in terms of (a) data volume and (b) encoding/decoding time (with baselines), required for CLIP ViT-B/32 to reach within 1% of peak performance on CIFAR-100. Data volume is normalized over full fine-tuning data size.

Table 1: Evaluation of **DeltaMask** using $Dir(10.0)$ over classes ($C_p \approx 1.0$ / IID settings) across architectures and pre-training strategies.

| Metric | CLIP ViT-B/32 | CLIP ViT-L/14 | DINOv2-Base | DINOv2-Small | ConvMixer-768/32 |
|---|---|---|---|---|---|
| Fine-tuning | $77.35 \pm 0.009$ | $89.07 \pm 0.012$ | $75.01 \pm 0.007$ | $65.55 \pm 0.019$ | $78.52 \pm 0.009$ |
| **DeltaMask** (Ours) | $75.82 \pm 0.023$ | $89.48 \pm 0.031$ | $73.36 \pm 0.027$ | $63.01 \pm 0.033$ | $75.31 \pm 0.021$ |
| Avg. bpp | $0.207 \pm 0.001$ | $0.225 \pm 0.002$ | $0.197 \pm 0.001$ | $0.214 \pm 0.001$ | $0.251 \pm 0.001$ |

include FedMask and FedPM, omitting their arithmetic encoding step to simplify computational complexity and ensure comparable execution times. All tests are conducted on a CPU, excluding aggregation in decoding time measurements. Data volume is normalized against full fine-tuning size, and we report the necessary volume to reach within 1% of peak accuracy, effectively combining communication efficiency with convergence speed analysis.

Among the evaluated methods in Figure 5, FedCode is the most communication-efficient in terms of data volume; yet, it suffers from longer encoding times and has the lowest model performance across baselines. Additionally, we notice that DeepReduce, utilizing a Bloom-based compressor, struggles with scalability due to longer execution times; in contrast, while **DeltaMask** offer significant improvements in filter construction and query times. FedMask and FedPM offer a compromise between data volume and execution time, with FedPM leading in accuracy among all approaches. Surprisingly, **DeltaMask**, while using slightly more data than FedCode, provides quicker encoding, critical for devices with limited resources, and matches the high accuracy of FedPM with significantly less data communicated. This positions **DeltaMask** as an effective choice for environments with computational and communication constraints. To further emphasize this point, we perform an analysis on multiple common edge devices utilized on edge in Appendix C.5.

## 5.4  Generalization across Neural Architectures and Pre-training Strategies

Next, we evaluate **DeltaMask** ability to work across various neural architectures pre-trained in a different self-supervised manner. We train masks for downstream task adaptation in a communication-constrained FL environment. For this, we perform experiments with $N=10$ on additional (larger) ViT architectures, namely CLIP-Large and DINOv2-Large, as well as a pure convolution-based architecture, ConvMixer-768/32 on CIFAR-100 as a downstream classification task. In all experiments, we mask the last 5 blocks, as discussed in Section 4. From Table 1, **DeltaMask** demonstrates robust adaptability across diverse pre-trained architectures in a FL setup with communication constraints. Notably, **DeltaMask** performance on large ViT architectures yield accuracies near those of fine-tuning, with CLIP ViT-L/14 slightly surpassing it. This is significant, considering the communication efficiency depicted by the average bitrate, which remains close to 0.2 bpp across all architectures. ConvMixer-768/32 also adapts well with **DeltaMask**, showing a modest accuracy reduction while meeting communication constraints. These results reinforce our method's suitability across

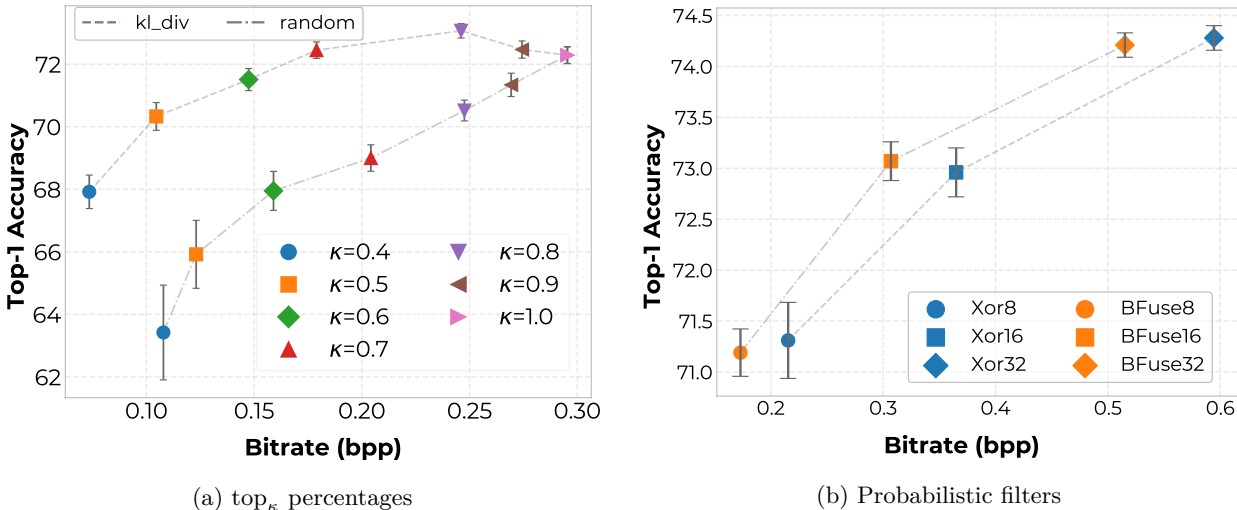

(a) $top_\kappa$ percentages

(b) Probabilistic filters

Figure 6: Impact of $top_\kappa$ mechanism and probabilistic filter choice in `DeltaMask` performance. Experiments performed in CIFAR-100 using Dir(10) over classes ($C_p \approx 1.0$ / IID settings). Federated parameters are set to $N$=10, $R$=100, $\rho$=1 and $E$=1.

diverse architectures, allowing for communication-efficient downstream task adaptation of FMs in a federated setting.

### 5.5 Adjusting Bitrate in `DeltaMask`

Now, we do ablation of fundamental components in `DeltaMask`: the mechanism sorting the positions of mask updates indexes and our choice of probabilistic filter, assessing the impact of these components on the model's final accuracy and bitrate. For this, we conduct experiments using CLIP ViT-B/32 with $N$=10 under full participation. In Figure 6a, we compare our entropy-based $top_\kappa$ sorting with a naive random sampling mechanism. We notice a consistent gap in performance between these approaches, underscoring the pivotal role of importance sampling in achieving generalization, similar to (Isik et al., 2023a). Surprisingly, increasing $\kappa$ does not linearly enhance accuracy, with the best results observed at $\kappa$=0.8, beyond which performance diminishes. This suggests that our $top_\kappa$ approach effectively filters out noise inherent in the stochastic binary mask sampling mechanism by prioritizing updates with higher certainty (i.e., higher probability), while also benefiting from a reduced bitrate due to transmitting less data.

Lastly, we evaluate the performance of various probabilistic filters, focusing on how variations in bits-per-entry (*bpe*), ranging from 8 to 32 bits, affect the false positive rate. Our analysis includes binary fuse filters (BFuse) and XoR (Graf & Lemire, 2020) filters, the latter operating under the same foundational principles but being slightly less space-efficient. Figure 6b demonstrates that BFuse filters generally surpass XoR filters in reducing bitrate without compromising model accuracy, a consistency observed across all experiments. More importantly, we demonstrate that `DeltaMask` enables an adjustable bitrate based on the *bpe* selection of the probabilistic filter, offering a potential solution to the resource heterogeneity among clients in FL.

## 6 Conclusions

We introduce `DeltaMask` a FL technique for efficiently fine-tuning FMs under low bitrate constraints by utilizing stochastic masking instead of conventional fine-tuning and leveraging probabilistic filters for communicating subsequent clients' mask updates. Our evaluation demonstrates `DeltaMask`'s effectiveness across a broad range of datasets and FMs, achieving significant communication reductions with performance similar to traditional fine-tuning. Beyond communication efficiency, `DeltaMask` can extend to offer personalized models in FL, while it can be expanded to adapt a single FM to multiple tasks, each with its uniquely learned masks. Additionally, the hashing operations of probabilistic filters and their false positive rate — interpreted

as a *bit-flipping* error in our masks — can potentially enhance privacy in FL. We believe this is a valuable directions for future research.

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

## A Bayesian Aggregation Algorithm

---
**Algorithm 2** BayesAgg
---
1: **Inputs:** Clients' updates $\{m_{k,t}\}_{k \in K}$, federated round $t$, and client participation $\rho$.
2: **Output:** Global probability mask $\theta_{g,t}$.
3: **if** $t \ \% \ (\frac{1}{\rho}) = 0$ **then**
4:     $\alpha_{g,t-1} = \beta_{g,t-1} = \lambda_0$
5: **end if**
6: $m^{\text{agg},t} \leftarrow \frac{1}{K} \sum_{k \in K} m^{k,t}$
7: $\alpha^{g,t} = \alpha^{g,t-1} + m^{\text{agg},t}$
8: $\beta^{g,t} = \beta^{g,t-1} + K \cdot 1 - m^{\text{agg},t}$
9: $\theta^{g,t} = \frac{\alpha^{g,t}}{\alpha^{g,t} + \beta^{g,t}}$
10: **return** $\theta^{g,t}$

---

## B Distributed Mean Estimation Error Analysis

We now provide proof of the upper bound on the estimation error of `DeltaMask`. Recall that we use probabilistic filters to reconstruct clients' binary masks, $m^{k,t} \sim \text{Bern}(\theta^{k,t})$ on server-side, which introduce an independent (across both clients and mask dimensions) "bit-flip" error probability $2^{-p}$ ($p$ referring to the false positive rate of the filter). We refer to these reconstructed masks as $m'^{k,t}$. Here, our true mean is $\bar{\theta}^{g,t} = \frac{1}{K} \sum_{k \in K_t} \theta_k^t$, while our estimate is $\hat{\bar{\theta}}^{g,t} = \frac{1}{K} \sum_{k \in K_t} m'^{k,t}$. Furthermore, we use capital letters to refer to random variables, while small letters refer to their deterministic quantities. We can then compute the error as follows:

$$\mathbb{E}_{M^{k,t} \sim \text{Bern}(\theta^{k,t}) \forall k \in K} \left[ \left\| \bar{\theta}^{g,t} - \hat{\bar{\theta}}^{g,t} \right\|_2^2 \right] = \sum_{i=1}^d \mathbb{E}_{M_i^{k,t} \sim \text{Bern}(\theta_i^{k,t}) \forall k \in K} \left[ \left( \bar{\theta}^{g,t} - \hat{\bar{\theta}}^{k,t} \right)^2 \right] \tag{7}$$

$$= \sum_{i=1}^d \mathbb{E}_{M_i^{k,t} \sim \text{Bern}(\theta_i^{k,t}) \forall k \in K} \left[ \left( \frac{1}{K} \sum_{k \in K} \left( M'^{k,t}_i - \theta_i^{k,t} \right) \right)^2 \right] \tag{8}$$

$$= \frac{1}{K^2} \sum_{i=1}^d \mathbb{E}_{M_i^{k,t} \sim \text{Bern}(\theta_i^{k,t}) \forall k \in K} \left[ \left( \sum_{k \in K} \left( M'^{k,t}_i - \theta_i^{k,t} \right) \right)^2 \right] \tag{9}$$

$$= \frac{1}{K^2} \sum_{i=1}^d \sum_{k \in K} \mathbb{E}_{M_i^{k,t} \sim \text{Bern}(\theta_i^{k,t})} \left[ \left( M'^{k,t}_i - \theta_i^{k,t} \right)^2 \right] \tag{10}$$

$$= \frac{1}{K^2} \sum_{i=1}^d \sum_{k \in K} \left( \mathbb{E}_{M_i^{k,t} \sim \text{Bern}(\theta_i^{k,t})} \left[ (M'^{k,t}_i)^2 \right] \right.$$
$$\left. -2\theta_i^{k,t} \mathbb{E}_{M_i^{k,t} \sim \text{Bern}(\theta_i^{k,t})} \left[ M'^{k,t}_i \right] + (\theta_i^{k,t})^2 \right) \tag{11}$$

$$= \frac{1}{K^2} \sum_{i=1}^d \sum_{k \in K} (\theta_i^{k,t} - (\theta_i^{k,t})^2) - 4 \cdot (2^{-p})(\theta_i^{k,t} - (\theta_i^{k,t})^2) + 2^{-p} \tag{12}$$

$$\leq \frac{d}{4K} \tag{13}$$

We begin by expressing the expected squared $L^2$ norm of the error $\hat{\bar{\theta}}^{g,t}$ and $\bar{\theta}^{g,t}$ in (7). From (7) to (8), we use the definitions of $\hat{\bar{\theta}}^{g,t} = \frac{1}{K} \sum_{k \in K_t} m'^{k,t}$ and $\bar{\theta}^{g,t} = \frac{1}{K} \sum_{k \in K_t} \theta_k^t$. To move from (9) to (10), we use the

fact that $M'^{k,t}$ and $M'^{l,t}$ are independent for $k \neq l$; thus the expected value of cross-product terms in the expansion of squared sum of is zero. At (11), we utilize the fact that $M'^{k,t}_i$ is a Bernoulli variable (meaning $(M'^{k,t}_i)^2 = M'^{k,t}_i$) and introduce the "*bit-flip*" error probability $2^{-p}$ due to the probabilistic filters; thus its expected value is $E[(M'^{k,t}_i] = (1 - 2^{-p})\theta^{k,t}_i + 2^{-p}\theta^{k,t}_i$. Finally, in (13), given that the variance of a Bernoulli random variable is maximized when the probability of success is 0.5, and that the flipping process does not change this maximum possible variance – as $2^{-p} << 1$ given $p \in \{8, 16, 32\}$ –we concluded that the upper bound of the expected squared error is $\frac{d}{4K}$, where $d$ is the number of dimensions and $K$ is the number of clients. It is important to note that our probabilistic filter-based encoding provides the same upper-bound estimation error as Isik et al. (2023b); yet, it can achieve significant reductions in terms of required bpp for transmitting masks.

## C   Additional Experiments

### C.1   Additional Experimental Details

**Training Parameters**: For our experiments, clients completed 1 local epoch per round with a batch size of 64 and Adam optimizer with a learning rate of 0.1. We adopted Bayesian aggregation, resetting the prior every $\frac{1}{\rho}$ rounds, where $\rho$ is the participation rate (as per Isik et al. (2023b)). In scenarios where $\rho$ is less than 1.0, client selection in each round was randomized. In most experiments, we set $\kappa$ to 0.8, except for those detailed in Figure 6a. We conducted 100 federated rounds for experiments with $\rho$=1 (both IID and non-IID settings) and increased the number of rounds to 200 and 300 for IID and non-IID experiments, respectively, when $\rho << 1$. Unless otherwise mentioned, we employed CLIP ViT-B/32 for experiments involving CLIP. We perform 3 independent runs and report the average accuracy on test-set in all our experiments.

**Baselines Configuration**: For FedMask, we set a binary threshold $\tau$ (masking with $m_i$=1 if $\theta_i \geq \tau$, and 0 otherwise) in the range [0.4, 0.6] for IID and [0.2, 0.0] for non-IID experiments, aligning with Isik et al. (2023b). In EDEN, a 1-bit gradient compression scheme was used to match the bitrate (*bpp*) of other baselines. Notably, EDEN's compression is model-dependent but yields nearly constant *bpp* reductions across all experiments. From DeepReduce compression, we discard the values' compression stage (as we deal we binary masks), and utilize only the Bloom filter-based index compression using the $P$0-policy Kostopoulou et al. (2021). Here, binary masks were learned via stochastic mask training (Isik et al. (2023b)), ensuring operation near the 1 *bpp* regime and facilitating clear comparison with `DeltaMask`. For our comparison with FedPM, we use identical settings albeit the compression scheme with probabilistic filters of `DeltaMask`, to clearly illustrate the benefits of our approach. We conducted our experiments on NVIDIA A10 GPUs on an internal cluster server, using 2 GPUs per one run.

### C.2   Additional Experiments in IID settings

In this section, we present additional experiments conducted under IID settings with varying participation rates ($\rho$). To ensure a fair comparison, we included both Linear Probing, which involves adapting a single linear classifier atop the (frozen) pre-trained model, and full Fine-tuning, wherein only the layers modified in `DeltaMask` are fine-tuned. In Table 2, apart from report models' accuracies across tasks, we include the average *bpp* and accuracy across all tasks for a concise comparison.

In Table 2, we note that `DeltaMask` achieves significant reductions in bitrate, while maintaining performance on par with Fine-tuning. This is particularly evident in scenarios with $\rho$ less than 1, where `DeltaMask` ability to reduce bitrate without compromising on accuracy highlights its effectiveness in federated learning environments with varying levels of client participation.

### C.3   Additional Experiments in non-IID settings

In this section, we provide additional experiments performed under non-IID settings, where we varied the participation rate ($\rho$). Similar to C.2, we include both Linear Probing and Fine-tuning for a rigorous

Table 2: Performance evaluation of **DeltaMask (Ours)** in terms of average bitrate (bits-per-parameter) during FL training using $Dir(10)$ over classes ($C_p \approx 1.0$ / **IID settings**) for CLIP ViT-B/32. Federated parameters are set to $N$=30 and $E$=1. For $\rho < 1$, clients are randomly selected.

| | Method | CIFAR-10 | CIFAR-100 | SVHN | EMNIST | Fashion-MNIST | EuroSAT | Food-101 | Cars196 | Avg. Acc | Avg. bpp |
|---|---|---|---|---|---|---|---|---|---|---|---|
| | Linear Probing | $92.12 \pm 0.007$ | $67.23 \pm 0.011$ | $59.70 \pm 0.016$ | $89.89 \pm 0.008$ | $89.05 \pm 0.010$ | $94.81 \pm 0.009$ | $67.58 \pm 0.014$ | $59.87 \pm 0.016$ | 77.51 | - |
| | Fine-tuning | $94.38 \pm 0.013$ | $76.12 \pm 0.019$ | $91.88 \pm 0.012$ | $94.02 \pm 0.018$ | $92.54 \pm 0.009$ | $97.61 \pm 0.015$ | $85.73 \pm 0.017$ | $66.98 \pm 0.011$ | **87.48** | 32 |
| $\rho = 0.2$ | FedMask | $85.32 \pm 0.033$ | $61.38 \pm 0.057$ | $68.71 \pm 0.046$ | $81.32 \pm 0.024$ | $84.32 \pm 0.044$ | $92.01 \pm 0.025$ | $62.28 \pm 0.037$ | $57.12 \pm 0.029$ | 74.06 | 1.0 |
| | EDEN | $87.11 \pm 0.006$ | $65.89 \pm 0.009$ | $79.16 \pm 0.008$ | $86.36 \pm 0.006$ | $85.21 \pm 0.012$ | $91.24 \pm 0.010$ | $69.59 \pm 0.012$ | $62.07 \pm 0.011$ | 78.33 | 0.703 |
| | DeepReduce | $86.71 \pm 0.071$ | $64.98 \pm 0.091$ | $60.32 \pm 0.061$ | $84.42 \pm 0.044$ | $84.09 \pm 0.057$ | $92.37 \pm 0.041$ | $64.91 \pm 0.043$ | $55.72 \pm 0.078$ | 73.61 | 1.123 |
| | FedPM | $90.31 \pm 0.016$ | $74.66 \pm 0.019$ | $87.03 \pm 0.017$ | $91.42 \pm 0.021$ | $89.79 \pm 0.013$ | $95.57 \pm 0.015$ | $74.80 \pm 0.014$ | $62.19 \pm 0.017$ | 83.22 | 0.946 |
| | DeltaMask | $89.52 \pm 0.021$ | $74.01 \pm 0.033$ | $86.86 \pm 0.024$ | $92.27 \pm 0.027$ | $89.68 \pm 0.014$ | $94.94 \pm 0.019$ | $74.09 \pm 0.029$ | $61.56 \pm 0.030$ | 82.87 | **0.197** |
| | Linear Probing | $93.97 \pm 0.004$ | $74.11 \pm 0.009$ | $59.26 \pm 0.011$ | $89.40 \pm 0.008$ | $89.47 \pm 0.005$ | $95.35 \pm 0.003$ | $76.64 \pm 0.009$ | $61.72 \pm 0.012$ | 79.99 | - |
| | Fine-tuning | $94.50 \pm 0.010$ | $77.35 \pm 0.009$ | $92.72 \pm 0.012$ | $94.89 \pm 0.010$ | $92.98 \pm 0.013$ | $98.24 \pm 0.011$ | $86.72 \pm 0.009$ | $67.23 \pm 0.014$ | **88.08** | 32 |
| $\rho = 1.0$ | FedMask | $90.84 \pm 0.028$ | $70.64 \pm 0.057$ | $74.32 \pm 0.039$ | $84.22 \pm 0.031$ | $88.64 \pm 0.029$ | $95.09 \pm 0.038$ | $68.46 \pm 0.034$ | $61.59 \pm 0.039$ | 79.23 | 1.0 |
| | EDEN | $93.15 \pm 0.009$ | $72.02 \pm 0.010$ | $86.67 \pm 0.007$ | $91.55 \pm 0.009$ | $90.40 \pm 0.012$ | $95.34 \pm 0.010$ | $80.02 \pm 0.004$ | $63.98 \pm 0.008$ | 84.14 | 0.691 |
| | DeepReduce | $88.17 \pm 0.034$ | $68.59 \pm 0.069$ | $62.34 \pm 0.056$ | $86.92 \pm 0.073$ | $85.44 \pm 0.031$ | $94.12 \pm 0.043$ | $67.92 \pm 0.075$ | $58.42 \pm 0.041$ | 76.53 | 1.089 |
| | FedPM | $93.58 \pm 0.014$ | $75.56 \pm 0.011$ | $88.76 \pm 0.013$ | $93.45 \pm 0.015$ | $92.10 \pm 0.009$ | $96.45 \pm 0.019$ | $83.45 \pm 0.013$ | $65.23 \pm 0.014$ | 86.07 | 0.872 |
| | DeltaMask | $93.50 \pm 0.019$ | $74.82 \pm 0.023$ | $87.95 \pm 0.021$ | $92.52 \pm 0.019$ | $91.27 \pm 0.023$ | $95.64 \pm 0.017$ | $82.73 \pm 0.024$ | $64.94 \pm 0.026$ | 85.44 | **0.151** |

evaluation. We report our findings in Table 3, where we also report the average *bpp* and accuracy across all tasks for a concise comparison of our baselines.

Table 3: Performance evaluation of **DeltaMask (Ours)** in terms of average bitrate (bits-per-parameter) during FL training using $Dir(0.1)$ over classes ($C_p \approx 0.2$ / **non-IID settings**) for CLIP ViT-B/32. Federated parameters are set to $N$=30 and $E$=1. For $\rho < 1$, clients are randomly selected.

| | Method | CIFAR-10 | CIFAR-100 | SVHN | EMNIST | Fashion-MNIST | EuroSAT | Food-101 | Cars196 | Avg. Acc | Avg. bpp |
|---|---|---|---|---|---|---|---|---|---|---|---|
| | Linear Probing | $84.51 \pm 0.019$ | $49.04 \pm 0.022$ | $43.16 \pm 0.020$ | $82.41 \pm 0.035$ | $86.29 \pm 0.024$ | $91.63 \pm 0.022$ | $51.54 \pm 0.021$ | $47.92 \pm 0.038$ | 67.06 | - |
| | Fine-tuning | $92.59 \pm 0.024$ | $70.20 \pm 0.037$ | $87.39 \pm 0.036$ | $92.00 \pm 0.057$ | $88.25 \pm 0.039$ | $95.56 \pm 0.029$ | $79.38 \pm 0.034$ | $60.11 \pm 0.051$ | **83.19** | 32 |
| $\rho = 0.2$ | FedMask | $83.14 \pm 0.059$ | $51.66 \pm 0.119$ | $51.78 \pm 0.049$ | $83.75 \pm 0.078$ | $85.91 \pm 0.073$ | $90.05 \pm 0.074$ | $53.19 \pm 0.063$ | $51.37 \pm 0.105$ | 68.86 | 1.0 |
| | EDEN | $87.87 \pm 0.037$ | $64.62 \pm 0.106$ | $81.06 \pm 0.081$ | $86.73 \pm 0.050$ | $86.75 \pm 0.055$ | $90.22 \pm 0.062$ | $72.55 \pm 0.056$ | $58.71 \pm 0.034$ | 78.56 | 0.715 |
| | DeepReduce | $86.07 \pm 0.097$ | $64.39 \pm 0.088$ | $82.92 \pm 0.071$ | $85.14 \pm 0.084$ | $83.91 \pm 0.067$ | $86.12 \pm 0.117$ | $52.92 \pm 0.055$ | $49.72 \pm 0.110$ | 73.90 | 1.173 |
| | FedPM | $90.70 \pm 0.045$ | $67.42 \pm 0.095$ | $87.51 \pm 0.079$ | $89.77 \pm 0.095$ | $88.42 \pm 0.092$ | $93.57 \pm 0.067$ | $76.80 \pm 0.076$ | $59.06 \pm 0.098$ | 81.64 | 0.948 |
| | DeltaMask | $90.32 \pm 0.083$ | $66.90 \pm 0.051$ | $87.36 \pm 0.093$ | $89.09 \pm 0.047$ | $86.91 \pm 0.067$ | $93.54 \pm 0.101$ | $76.39 \pm 0.086$ | $58.52 \pm 0.102$ | 81.13 | **0.233** |
| | Linear Probing | $91.46 \pm 0.026$ | $71.96 \pm 0.025$ | $46.03 \pm 0.017$ | $84.57 \pm 0.031$ | $87.13 \pm 0.018$ | $92.98 \pm 0.016$ | $68.70 \pm 0.028$ | $54.03 \pm 0.032$ | 74.61 | - |
| | Fine-tuning | $93.61 \pm 0.048$ | $75.49 \pm 0.052$ | $90.10 \pm 0.063$ | $93.13 \pm 0.037$ | $91.06 \pm 0.041$ | $97.02 \pm 0.034$ | $84.71 \pm 0.013$ | $64.93 \pm 0.063$ | **86.26** | 32 |
| $\rho = 1.0$ | FedMask | $88.42 \pm 0.051$ | $63.04 \pm 0.081$ | $64.32 \pm 0.073$ | $86.41 \pm 0.039$ | $86.39 \pm 0.044$ | $91.67 \pm 0.031$ | $68.04 \pm 0.050$ | $54.39 \pm 0.089$ | 75.34 | 1.0 |
| | EDEN | $92.14 \pm 0.043$ | $71.65 \pm 0.060$ | $86.28 \pm 0.057$ | $90.87 \pm 0.046$ | $89.94 \pm 0.034$ | $93.26 \pm 0.035$ | $78.79 \pm 0.083$ | $61.18 \pm 0.027$ | 83.01 | 0.703 |
| | DeepReduce | $87.33 \pm 0.052$ | $67.19 \pm 0.061$ | $83.19 \pm 0.048$ | $85.71 \pm 0.082$ | $84.52 \pm 0.075$ | $92.12 \pm 0.060$ | $69.11 \pm 0.092$ | $60.31 \pm 0.094$ | 78.69 | 1.092 |
| | FedPM | $92.99 \pm 0.045$ | $74.34 \pm 0.023$ | $89.35 \pm 0.025$ | $92.65 \pm 0.098$ | $91.33 \pm 0.041$ | $95.37 \pm 0.048$ | $83.69 \pm 0.076$ | $63.65 \pm 0.074$ | 85.42 | 0.901 |
| | DeltaMask | $92.84 \pm 0.083$ | $73.69 \pm 0.051$ | $89.01 \pm 0.085$ | $91.92 \pm 0.089$ | $91.27 \pm 0.055$ | $94.54 \pm 0.103$ | $83.48 \pm 0.081$ | $63.47 \pm 0.096$ | 85.03 | **0.191** |

Table 3 reveals a notable improvement in `DeltaMask` performance, especially when the participation ratio $\rho$ is less than 1, with only a 2% accuracy difference compared to Fine-tuning. This is a critical observation, since non-IID data distributions coupled with partial client participation closely mirror the conditions of real-world federated settings. Furthermore, our analysis shows that methods using stochastic mask training, such as DeepReduce and FedPM, yield better final model accuracy under non-IID conditions than traditional compression schemes like EDEN or hard-thresholding masking techniques like FedMask. Interestingly, the CLIP ViT-B/32 model excels in non-IID scenarios, underscoring the robust generalization abilities of pre-trained foundation models, which are particularly advantageous in non-IID federated environments. This emphasizes the importance of adapting these models for edge computing, capitalizing on their capability to effectively handle diverse and complex data distributions.

## C.4 Experiments in ImageNet datasets

In this section, we extend our evaluation in more complex tasks to assess the effectiveness of `DeltaMask` to fine-tune FMs in federated settings in a communication-efficient manner under datasets of larger complexity. For this, we perform experiments on Tiny-ImageNet Le & Yang (2015) with both CLIP ViT-B/32 and CLIP ViT-L/14. The results, reported on Table 4 showcase that `DeltaMask` can effectively fine-tune FMs in more complex tasks, such as ImageNet datasets, while maintaining the same efficiency in terms of *bpp*.

Table 4: Evaluation of `DeltaMask` using $Dir(10.0)$ over classes ($C_p \approx 1.0$ / IID settings) across CLIP architectures in Tiny-ImageNet Le & Yang (2015).

| Method | CLIP ViT-B/32 | | CLIP ViT-L/14 | |
|---|---|---|---|---|
| | Accuracy | Avg. bpp | Accuracy | Avg. bpp |
| Fine-tuning | 86.12 | 32 | 89.02 | 32 |
| FedPM | 84.22 | 0.871 | 87.04 | 0.862 |
| DeltaMask (Ours) | 83.76 | **0.201** | 86.57 | **0.218** |

## C.5 `DeltaMask` Efficiency on Edge Devices

In this section, we evaluate the runtime resource demands—computation and energy—of our probabilistic filter compression on three popular embedded platforms: NVIDIA Jetson Nano (4GB), Raspberry Pi 4 (4GB), and Coral Dev Board (1GB). These platforms were selected for their widespread use and capability to run machine learning tasks at the edge. To measure energy consumption, we used a Raspberry Pi 4 equipped with a Current/Power Monitor HAT, monitoring each device's energy use with a 0.1 Ohm sampling resistor. Our tests, conducted over 5 runs, record the average runtime (in milliseconds) and energy usage (in nano Joules) for different probabilistic filters with varying bits per entry (8, 16, and 32), as detailed in Table 5.

Table 5: Average energy and latency benchmarking of the considered probabilistic filters across different devices. The CPU execution time (ms) and estimated energy consumption (nJ) per entry is computed over 10M entries.

| Filter | Metric | Raspberry Pi 4 | Coral Dev Board | Jetson Nano |
|---|---|---|---|---|
| Xor8 | CPU Execution Time (ms) | $0.942 \pm 0.0165$ | $1.682 \pm 0.0059$ | $0.479 \pm 0.0001$ |
| | Energy Consumption (nJ) | $3.223 \pm 0.0023$ | $2.826 \pm 0.0011$ | $2.334 \pm 0.0012$ |
| Xor16 | CPU Execution Time (ms) | $0.955 \pm 0.0250$ | $1.683 \pm 0.0008$ | $0.502 \pm 0.0001$ |
| | Energy Consumption (nJ) | $4.052 \pm 0.0032$ | $3.580 \pm 0.0003$ | $3.386 \pm 0.0016$ |
| Xor32 | CPU Execution Time (ms) | $0.978 \pm 0.0278$ | $1.701 \pm 0.0006$ | $0.539 \pm 0.0005$ |
| | Energy Consumption (nJ) | $6.292 \pm 0.0021$ | $4.732 \pm 0.0008$ | $4.692 \pm 0.0023$ |
| BFuse8 | CPU Execution Time (ms) | $\mathbf{0.587 \pm 0.0059}$ | $\mathbf{1.144 \pm 0.0035}$ | $\mathbf{0.289 \pm 0.0013}$ |
| | Energy Consumption (nJ) | $\mathbf{2.045 \pm 0.0019}$ | $\mathbf{1.979 \pm 0.0015}$ | $\mathbf{1.829 \pm 0.0023}$ |
| BFuse16 | CPU Execution Time (ms) | $0.590 \pm 0.0066$ | $1.183 \pm 0.0029$ | $0.282 \pm 0.0002$ |
| | Energy Consumption (nJ) | $3.262 \pm 0.0020$ | $2.898 \pm 0.0017$ | $2.157 \pm 0.0033$ |
| BFuse32 | CPU Execution Time (ms) | $0.612 \pm 0.0054$ | $1.201 \pm 0.0017$ | $0.301 \pm 0.0002$ |
| | Energy Consumption (nJ) | $4.021 \pm 0.0026$ | $3.771 \pm 0.0022$ | $3.263 \pm 0.0012$ |

From the results, we clearly notice that all filter variants demand limited computational resources, both in terms of execution time and energy requirements. `BFuse8` is particularly notable for its efficiency, requiring only an average execution time of 0.673 milliseconds and consuming just 1.95 nano Joules of energy across the considered devices. This underscores the practicality of our probabilistic filter-based compression scheme in federated settings, where devices are often constrained by limited computational capabilities and strict energy budgets. Additionally, our analysis shows that even with an increase in the bits-per-entry (*bpe*) parameter, the rise in execution time and energy consumption is quite modest. This is particularly noteworthy given the simultaneous improvement in the false positive rate, which is inversely proportional to $2^{-\text{bpe}}$. This pattern suggests a beneficial trade-off between accuracy and resource utilization, reinforcing the adaptability and effectiveness of our approach in federated learning scenarios that prioritize computational efficiency and energy conservation.

## C.6 Comparing Classifier Heads in `DeltaMask`

In `DeltaMask`, we enable the classification head to adapt in a single linear probing round, while freezing the rest of the model. This approach produces more stable outcomes and quicker convergence than previous methods Isik et al. (2023b); Zhou et al. (2020); Ramanujan et al. (2020); Zhou et al. (2020) that used Kaiming initialization to identify high-performing subnetworks in randomly initialized networks. Although

the classification head typically has fewer parameters, scenarios requiring extremely low bitrates make transmitting even a single round's floating-point weights impractical. In this section, we explore such situations, investigating different alternatives for the classifier layer. Specifically, we replace the linear classifier with a Gaussian Naive Bayes classifier from FiT Shysheya et al. (2022), specifically *FIT-LDA*. This classifier is data-driven, with a minimal number of learnable parameters (2 float-point values), making it ideal for our purpose. In our analysis, we utilize CLIP ViT-B/32, masking the last five transformer blocks and compare `DeltaMask`$_{\text{FiT}}$ against both a single-round trained linear classifier (`DeltaMask`$_{\text{LP}}$) and a Kaiming initialized (frozen) classifier (`DeltaMask`$_{\text{He}}$).

Table 6: Evaluating Classifier Initialization Schemes in `DeltaMask`. Comparing Average Bitrate and Accuracy in FL Training using $Dir(10)$ over classes ($C_p \approx 1.0$ / **IID settings**) for CLIP ViT-B/32. Federated parameters are set to $N=30$ and $E=1$.

| Method | CIFAR-10 | CIFAR-100 | SVHN | EMNIST | Fashion-MNIST | EuroSAT | Food-101 | Cars196 | Avg. Acc | Avg. bpp |
|---|---|---|---|---|---|---|---|---|---|---|
| Fine-tuning | $94.50 \pm 0.010$ | $77.35 \pm 0.009$ | $92.72 \pm 0.012$ | $94.89 \pm 0.010$ | $92.98 \pm 0.013$ | $98.24 \pm 0.011$ | $86.72 \pm 0.009$ | $67.23 \pm 0.014$ | 88.08 | 32 |
| `DeltaMask`$_{\text{He}}$ | $90.28 \pm 0.052$ | $67.34 \pm 0.069$ | $84.09 \pm 0.063$ | $87.32 \pm 0.081$ | $87.69 \pm 0.034$ | $93.22 \pm 0.073$ | $78.05 \pm 0.028$ | $58.74 \pm 0.084$ | 80.84 | **0.143** |
| `DeltaMask`$_{\text{FiT}}$ | $93.42 \pm 0.023$ | $71.17 \pm 0.041$ | $86.31 \pm 0.039$ | $92.09 \pm 0.021$ | $89.87 \pm 0.026$ | $95.53 \pm 0.019$ | $81.71 \pm 0.033$ | $60.01 \pm 0.029$ | 83.76 | 0.145 |
| `DeltaMask`$_{\text{LP}}$ | $93.50 \pm 0.019$ | $74.82 \pm 0.023$ | $87.95 \pm 0.021$ | $92.52 \pm 0.019$ | $91.27 \pm 0.023$ | $95.64 \pm 0.017$ | $82.73 \pm 0.024$ | $64.94 \pm 0.026$ | **85.44** | 0.151 |

From Table 6, we notice that `DeltaMask`$_{\text{LP}}$ outperforms other initialization methods by over 2% without significantly increasing the bitrate, while *FiT* can be an effective alternative to Kaiming initialization, increasing accuracy by $\approx$3%. More importantly, these findings highlight the importance of appropriate classifier layer initialization during fine-tuning of foundation models in downstream tasks. However, we demonstrate that a single fine-tuning round of the classifier layer, with the remaining model frozen, is an effective strategy with minimal impact on the communicated bitrate.

