# OpenReview forum: "Federated Fine-Tuning of Vision Foundation Models via Probabilistic Masking"
_TMLR — Rejected by TMLR_

### Review · Reviewer_yNj8 · 2024-05-24

**Summary Of Contributions:**

This work proposes a compression mechanism for federated learning communication in the pretrained foundation model setting (e.g, ViTs). The mechanism, DeltaMask, is based off of a combination of multiple techniques, namely masking, delta compression, and importance sampling. The updates are transmitted using probabilistic data structures as well as PNG image compression. Evaluations are over 8 datasets and 5 pretrained models.

**Audience:**

Yes

**Claims And Evidence:**

Yes

**Requested Changes:**

Critical:

* Can you please differentiate this work with prior work in introduction? For example, how does the work compare with the cited work of Isik et. al. 2023?

Suggested:

* Can you please motivate some of the design choices up front, such as why PNG compression and/or a particular probabilistic data structure is used? It is difficult to understand what are fundamental ideas and what are engineering tradeoffs.

**Strengths And Weaknesses:**

Strengths:
* Writing is clear.
* The method seems strong in the Pareto sense, particularly in the low bit regime.

Weaknesses:
* The method is heavily engineered with many components, thus may be more difficult to adapt and understand.
* Many of the design choices aren't motivated up front. For example, PNG compression is mentioned as a way to transmit updates, but it's not clear why this is preferable to other lossless algorithms. For similar reasons, it's not clear what techniques are different from prior work.

---

> ### Author Response · Authors · 2024-06-06
> **Reply to Reviewer yNj8 (1/2)**
>
> We appreciate the reviewer's positive feedback regarding the clarity of our writing and the effectiveness of DeltaMask, particularly in extremely low bitrate regimes. To address the reviewer's concerns and enhance the manuscript's clarity, we have made several changes and highlighted these in blue in our revision.
>
> While our work primarily demonstrated that DeltaMask, combined with the probabilistic mask training of Isik et al. 2023b, can significantly reduce communication overhead in fine-tuning vFMs in FL, it can also be adapted to other probabilistic training regimes that rely on binary mask updates without requiring any modifications. Furthermore, we have revised our manuscript to better highlight our design choices and clarify how DeltaMask differs from prior work. Detailed responses to the reviewer's suggestions regarding DeltaMask’s novelty and rationale of design choices are provided below.
>
>
> > *Can you please differentiate this work with prior work in introduction? For example, how does the work compare with the cited work of Isik et. al. 2023?*
> >
>
> We appreciate the reviewer’s request for clearer differentiation from prior work in our introduction. Isik et al. 2023b explore probabilistic mask training in federated learning (FL) to achieve communication efficiency with a bitrate of approximately 1 bpp, focusing on training smaller, randomly initialized models from scratch. In contrast, DeltaMask is developed for fine-tuning pre-trained vision Foundation Models (vFMs) in FL that can have extensive parametrization, leading to distinct challenges, as even 1 bpp necessitates significant communication costs to adapt vFMs to new tasks.
>
> We capitalize on a key insight in the development of DeltaMask: fine-tuning vFMs through probabilistic masking primarily results in a sparse number of bit-flips between mask transmissions. This observation led us to develop a simple yet effective two-step mask compression scheme that combines binary fuse filters with lossless image compression. As a result, DeltaMask efficiently encodes bit-flip positions and reduces redundancies, achieving extremely low bitrates — well below 1 bpp — while maintaining the same upper bound estimation error as Isik et al. 2023b. Our approach to encoding with probabilistic filters simplifies the control over the bitrate-accuracy trade-off by adjusting the bits-per-element (bpe) in binary fuse filters, as shown in Fig.6b. This method eliminates the need for the complex synchronization schemes required by Isik et al. 2023a, and it introduces no communication overhead.
>
> Furthermore, while mask training is a recognized technique within the pruning literature, its application for achieving ultra-low bitrates in a communication-efficient manner in pre-trained vFMs has been largely unexplored. DeltaMask not only addresses this gap but also demonstrates the practical feasibility of conducting such advanced mask training within FL frameworks.

---

> > ### Author Response · Authors · 2024-06-06
> > **Reply to Reviewer yNj8 (2/2)**
> >
> > > *Can you please motivate some of the design choices up front, such as why PNG compression and/or a particular probabilistic data structure is used? It is difficult to understand what are fundamental ideas and what are engineering tradeoffs.*
> >
> > We recognize the importance of clearly distinguishing between fundamental principles and engineering considerations in DeltaMask.
> >
> > * **PNG Compression**: The selection of PNG compression for transmitting hashed BFuse filter entries in DeltaMask was primarily driven by practical considerations, particularly its widespread optimization on edge devices commonly used in FL environments. This includes both the efficiency of the compression algorithm itself and the effectiveness of associated transmission techniques. While other lossless image compression schemes could potentially be employed and were not explored in this work, it is crucial to note that the primary source of bitrate reduction in DeltaMask stems from earlier steps, namely delta compression and BFuse filter-based encoding of indices.
> > * **Binary Fuse Filters**: The choice of binary fuse filters as our probabilistic data structure in DeltaMask was primarily motivated by their exceptional balance between space efficiency and false positive rate, as detailed in Graf & Lemire (2022). Their superiority over other probabilistic data structures, such as Bloom or XoR filters, is clearly evident in our evaluations (see Table 5). When compared to DeepReduce, DeltaMask consistently shows a significant accuracy advantage across all experiments. Additionally, these filters provide considerable computational benefits, crucial for the resource-constrained environments typically found on edge devices (refer to Table 5 for experiments on popular edge devices).
> >
> > We have added these clarifications regarding our design choices in our revised manuscript.
> >
> > [1]: E. Öztürk and A. Mesut, "Performance Evaluation of JPEG Standards, WebP and PNG in Terms of Compression Ratio and Time for Lossless Encoding," 2021 6th International Conference on Computer Science and Engineering (UBMK), Ankara, Turkey, 2021.

---

### Review · Reviewer_B1Gi · 2024-05-30

**Summary Of Contributions:**

Integration of Large Foundation Models into Federated Learning (FL) is challenging due to substantial communication overhead.

Current communication efficient FL strategies (e.g., gradient compression) reduce bitrates to around 1 bit-per-parameter (bpp). But, it fail to harness the characteristics of FM, with their large number of parameters still posing challenge to communicate efficiently, even at these bitrate regime.

The work present DeltaMask that fine-tunes FMs in FL at an ultra-low bitrate, well below 1 bpp. It uses stochastic masking to detect highly effective subnetworks within the model which is leveraged for efficient finetuning.

**Audience:**

Yes

**Broader Impact Concerns:**

Not applicable.

**Claims And Evidence:**

Yes

**Requested Changes:**

See Weaknesses above.

**Strengths And Weaknesses:**

[Strengths]
* The proposed method shows a clear advantage in terms of bitrate (bpp) while having comparable accuracy to previous methods.

* Extensive experiments on various datasets and architectures.

* The paper is well-organized with clear writing.

[Weaknesses]
* typo: page 8 line 12: DeltaMaskagainst → DeltaMask against.
* citation format: The citation should be inside the parenthesis. For example, Method A (~~~ et al.)

---

> ### Author Response · Authors · 2024-06-06
> **Reply to Reviewer B1Gi**
>
> We would like to thank the reviewer for their encouraging comments on the clarity of our manuscript and the efficiency of DeltaMask at extremely low bitrates. In our revised manuscript, we have corrected all typos and adjusted the citation style as per your suggestion to further improve our work.

---

### Review · Reviewer_3HHk · 2024-06-02

**Summary Of Contributions:**

This paper studies the federated fine-tuning of foundation models, in particular, focusing on the design of fine-tuning algorithms that operate at low bit rates. The motivation for this constraint is to reduce the communication bottleneck of training federated learning systems.

The main content of the paper can be summarized as follows.
- In section 3, the authors discuss the methodology, first introducing what probabilistic filters are---These use hash functions to transform and store data in a uniformly distributed array, known as the fingerprints. The stochastic mask training creates binary masks by clipping mask scores, drawing from an underlying mask probability using the Bernoulli distribution.
    - The federated masked fine-tuning is then presented. Here, clients first initialize a neural network with a weight vector from the pre-trained foundation model. The weight vector is kept fixed and never modified during training. The algorithm collaboratively trains a probabilistic mask and the loss function to minimize its error rate on a downstream task.
    - Instead of communicating the stochastic binary mask, the paper proposes to reduce the required bits-per-parameter by solely communicating the subsequent key updates between the received and trained mask.
    - The paper utilizes a local training scheme for probability masks. It also leverages the inherent sparsity in mask updates during federated rounds.
    - Once the local training is completed, the server forms a new global probability mask through a Bayesian aggregation of the compressed models. In contrast to related approaches, the method enables better control over model generalization.
- In sections 4 and 5, the authors introduce experiments. discussing
    - Bit rate vs. accurate trade-off
    - Data volume and computational cost improvements
    - The algorithm's ability to work across various architectures
    - Ablation studies including the performance of various probabilistic filters.

**Audience:**

Yes

**Claims And Evidence:**

No

**Requested Changes:**

- The first requested change is to fix the issues discussed under the weaknesses above. However, to me this might require a complete re-write of section 3 and a large part of section 4/5, which would be a major revision of the manuscript.

- The second requested change is to clarify the claims on the Privacy Implications for FL. This seems like a major weakness to me as the authors are claiming a FL training algorithm that might protect user data privacy. However, the paper makes no such claims in the paper. This is a major omission---One possible fix is to add some justification based on the Differential Privacy framework (see, e.g., "The Algorithmic Foundations of Differential Privacy" by Dwork and Roth). However, this again requires a major revision of the manuscript.

Taking both issues above, I thus feel that this paper is not yet ready for publication, although the problem is well-motivated.

**Strengths And Weaknesses:**

Strengths: The paper studies a well-motivated problem, discussing efficient fine-tuning algorithms in the federated learning setting. The idea of communicating probabilistic masks and compressing that appears to be a simple, yet effective idea. The comparisons between the proposed algorithm (DeltaMask) and baeslines in terms of the bit rates appear to be interesting.

Weaknesses:
- In section 3, I have found the methodology section to be rather scattered and poorly written.
    - for instance, going from section 3.1 to section 3.2 was a jump: the previous section taslks about masking schemes. While the next gives an overview of the pipeline. However, it's not clear to me what part of section 3.1 is being used in section 3.2.
    - in section 3.2, the authors claim that "the method enables better control over generalization compared to related approaches." However, I find the arguments to be unconvincing. I think more analysis needs to be done here before making this claim. Also, the readers did not get a preview of FedPM and HidNseek, making the claim difficult to verify.
    - complete description of the algorithm needs to be in the main text. Otherwise, I could not tell what exactly is the method being implemented.
- In section 4, this should be organized more coherently and the baselines should be described more clearly, justifying why these are sufficient for the authors to draw a conclusion.
- In section 5, again the authors should list a summary of the main empirical findings here. There are much details mixed with description of the results in this section.
    - In figure 5 / table 1, it is unclear to me why these results are significant. Also, the rationale / intuition for the comparison is not clearly explained.

---

> ### Author Response · Authors · 2024-06-06
> **Reply to Reviewer 3HHk (1/2)**
>
> We would like to thank the reviewer for the thorough review of our manuscript. We appreciate your insightful comments and suggestions, which have greatly contributed to improving the clarity and quality of our work. We have made revisions to address your concerns, which are highlighted in blue in the revised manuscript.
>
>
> > *In section 3, the authors claim that "the method enables better control over generalization compared to related approaches." However, I find the arguments to be unconvincing. I think more analysis needs to be done here before making this claim. Also, the readers did not get a preview of FedPM and HidNseek, making the claim difficult to verify.*
>
>
> We thank the reviewer's suggestion to improve the readability of our manuscript. In response, we have adapted the related work section to explicitly provide an overview of FedPM and HideNseek. We would like to point to the reviewer that our claim is that DeltaMask allows for better control over bitrate versus model generalization trade-off, and not regarding model generalization in general. By controlling $\kappa$ parameters and bpe of probabilistic filter, DeltaMask allows for a finer-grained control over this trade-off, as evident in Figure 6. In contrast, HideNseek simply transmits raw mask updates, whereas FedPM utilizes arithmetic encoding over a client’s mask to reduce the bitrate.Therefore, these approaches control over the bitrate versus model generalization trade-off is limited to the effectiveness of arithmetic encoding schemes (i.e., the current mask structure). However, we are the first to exploit the sparsity of subsequent mask updates, enabling fine-tuning of pre-trained vFMs under extremely low bitrates. We have revised our manuscript to clarify our claims in the revision.
>
> > *In section 3, I have found the methodology section to be rather scattered and poorly written. For instance, going from section 3.1 to section 3.2 was a jump: the previous section talks about masking schemes. While the next gives an overview of the pipeline. However, it's not clear to me what part of section 3.1 is being used in section 3.2.*
>
> We have made changes in our original manuscript to improve the clarity of Section 3. Specifically, we have added an introduction to Section 3.1 to indicate how these fundamental concepts are used in our work, providing an early glimpse of our methodology to the reader. To further enhance this point, we have adjusted the introduction of Section 3.2 and replaced the “Overview” subsection of Section 3.2, with an introduction section that provides the reader with direct links to the formulas introduced in Section 3.1 and how they are applied in our approach (please see first two paragraphs of Section 3.2).
>
> > *Complete description of the algorithm needs to be in the main text. Otherwise, I could not tell what exactly is the method being implemented.*
>
> We thank the reviewer for their suggestion. In response, we have included a complete description of the algorithm in the main manuscript. We believe this addition will help readers have a better understanding of our work.
>
> > *In section 4, this should be organized more coherently and the baselines should be described more clearly, justifying why these are sufficient for the authors to draw a conclusion.*
>
> We have removed redundancies and reformatted Section 4 for better coherence. Additionally, we have adapted our baselines text to justify our choices and explain how these help draw conclusions about our work's superiority in low bitrate regimes (see “Baselines” in Section 4).
>
> >  *In section 5, again the authors should list a summary of the main empirical findings here. There are many details mixed with description of the results in this section.*
>
> We would like to thank the reviewer for their suggestion. Our main empirical findings are now summarized more clearly in the beginning of Section 5. These include significant bit rate reductions, accuracy per total bit transmitted, computational efficiency, model-agnostic performance, and control over the model generalization versus bitrate trade-off (please see Section 5.1).

---

> > ### Author Response · Authors · 2024-06-06
> > **Reply to Reviewer 3HHk (2/2)**
> >
> > > *In figure 5 / table 1, it is unclear to me why these results are significant. Also, the rationale / intuition for the comparison is not clearly explained.*
> >
> > In Figure 5a, we aim to compare the total size of communicated data with related works, highlighting DeltaMask's advantages under strict data transmission limits. DeltaMask achieves the highest accuracy per total bit transmitted compared to all considered methods. Furthermore, Figure 5b shows that the computational efficiency rivals traditional gradient compression schemes, exhibiting similar encode and decode times. This is important, especially considering that in FL, devices often operate under strict energy and compute budgets and our scheme does not introduce a computational bottleneck on the server-side due to probabilistic filters’ membership checks (see also Table 5). In Table 1, we demonstrate that our approach is model-agnostic and effective across various architectures, with DeltaMask performing exceptionally well with large models, such as ViT-L/14, surpassing even traditional fine-tuning. We have added these clarifications in our revised manuscript.
> >
> > > *The second requested change is to clarify the claims on the Privacy Implications for FL. This seems like a major weakness to me as the authors are claiming a FL training algorithm that might protect user data privacy. However, the paper makes no such claims in the paper. This is a major omission---One possible fix is to add some justification based on the Differential Privacy framework (see, e.g., "The Algorithmic Foundations of Differential Privacy" by Dwork and Roth). However, this again requires a major revision of the manuscript.*
> >
> > We agree that our work is not focused on privacy and should not be presented as such. Instead, we pointed out that our encoding scheme has potential advantages in privacy preservation, which can be a promising future direction. We have moved this discussion to the conclusions section to highlight it as an important future work (see Section 6).

---

### Decision · Action_Editor_2sKX · 2024-08-02

**Recommendation:** Reject

**Comment:**

This paper considers the problem of fine-tuning models in a communication-constrained federated learning setting. The parameters of a pre-trained model are shared with client devices. For the fine-tuning process, DeltaMask is proposed, a technique where clients collaborate in training a stochastic binary mask. DeltaMask leverages stochasticity and sparsity in client masks to compress updates and allows fine-tuning at bit-per-parameter rates below 1 (previous binary mask-based procedures used 1 bit per parameter). Instead of communicating the stochastic binary mask, the paper proposes to reduce the required bits-per-parameter by solely communicating the subsequent key updates between the received and trained mask.

The reviewers found that the proposed method has a better bitrate (bpp) while being comparable to previous methods in accuracy. However, they also noted that the approach is rather complex regarding engineering detail and, therefore, is perhaps more challenging to implement, verify, and generalize.

The review committee unanimously finds that this paper presents strong and promising results but offers limited insights and reproducibility. For example, the main points raised by the reviewers is that:
- the control of the bitrate in the proposed DeltaMask is vaguely described (the paper mentions control of $\kappa$ and bpe, but these do not appear as parameters in Algorithm 1)
- is challenging to understand what fundamental ideas are and what engineering tradeoffs are

As the reviewers did not reach a consensus on this work (with none arguing strongly for acceptance), I also looked closely at the paper before proposing a decision. I mainly checked if the paper's description of the approach is clear enough to verify the claims according to TMLR's evaluation criteria. However, I find that the current version of the manuscript does not entirely meet these expectations and that a revision would be incredibly beneficial to the readers and the TMLR audience.

For example, Section 3.1 is very dense and introduces many parameters ($\mu$, $l$, $n$, $m$, $\mathcal{U}$, $\mathcal{H}$) but the effect of these parameters is not well-explained later (e.g. how to achieve the desired adjustments in bpp, etc.). Moreover, it is additionally confusing that the notation does not seem consistent with the other sections ($m$ for mask, $k$ for client index, etc.).

For a potential revision submitted to TMLR, we suggest separating the machine learning components from the engineering aspects to enhance the clarity of the presentation. To better align with the expectations of the TMLR audience and reviewers, please also consider the following points the reviewers raised:
(A) a clean and detailed presentation of the fine-tuning pipeline with masking, highlighting how it contrasts with other fine-tuning approaches in machine learning.
(B) more intuition and explanations behind the main idea (trade-offs between compressing the updates vs. compressing masks/parameters)
(C) a thorough description of DeltaMask, including detailed information on hyperparameters and their effects.

Addressing these points could significantly improve the submission.

**Audience:**

The paper deals with a well-motivated problem in federated learning using large models. The proposed solution could be of interest to the TMLR community.

**Claims And Evidence:**

The reviewing committee did not reach a consensus on this question. The primary reason is that the proposed pipeline appears complex, and the current version of the manuscript makes the claims difficult to verify. An example mentioned by reviewers is the claim that DeltaMask enables effective control of the bitrate, but it is unclear how the hyperparameters would need to be adjusted precisely. We believe a revision could clarify the reviewers' concerns and benefit the readers.

**Resubmission Of Major Revision:**

The authors may consider submitting a major revision at a later time.